# Learning to Predict 3D Objects with an Interpolation-based Differentiable Renderer

Wenzheng Chen[1,2,3]    Jun Gao[1,2,3,*]    Huan Ling[1,2,3,*]    Edward J. Smith[1,4,*]

Jaakko Lehtinen[1,5]    Alec Jacobson[2]    Sanja Fidler[1,2,3]

NVIDIA[1]    University of Toronto[2]    Vector Institute[3]    McGill University[4]    Aalto University[5]

{wenzchen, huling, jung, esmith, jlehtinen, sfidler}@nvidia.com, jacobson@cs.toronto.edu

## Abstract

Many machine learning models operate on images, but ignore the fact that images are 2D projections formed by 3D geometry interacting with light, in a process called rendering. Enabling ML models to understand image formation might be key for generalization. However, due to an essential rasterization step involving discrete assignment operations, rendering pipelines are non-differentiable and thus largely inaccessible to gradient-based ML techniques. In this paper, we present *DIB-R*, a differentiable rendering framework which allows gradients to be analytically computed for all pixels in an image. Key to our approach is to view foreground rasterization as a weighted interpolation of local properties and background rasterization as a distance-based aggregation of global geometry. Our approach allows for accurate optimization over vertex positions, colors, normals, light directions and texture coordinates through a variety of lighting models. We showcase our approach in two ML applications: single-image 3D object prediction, and 3D textured object generation, both trained using exclusively using 2D supervision. Our project website is: https://nv-tlabs.github.io/DIB-R/

## 1   Introduction

3D visual perception contributes invaluable information when understanding and interacting with the real world. However, the raw sensory input to both human and machine visual processing streams are 2D projections (images), formed by the complex interactions of 3D geometry with light. Enabling machine learning models to understand the image formation process could facilitate disentanglement of geometry from the lighting effects, which is key in achieving invariance and robustness.

The process of generating a 2D image from a 3D model is called *rendering*. Rendering is a well understood process in graphics with different algorithms developed over the years. Making these pipelines amenable to deep learning requires us to differentiate through them.

In [17], the authors introduced a differentiable ray tracer which builds on Monte Carlo ray tracing, and can thus deal with secondary lighting effects such as shadows and indirect light. Most of the existing work focuses on rasterization-based renderers, which, while simpler in nature as they geometrically project 3D objects onto the image plane and cannot support more advanced lighting effects, have been demonstrated to work well in a variety of ML applications such as single-image 3D prediction [22, 14, 20, 21]. Here, we follow this line of work.

Existing rasterization-based approaches typically compute approximate gradients [22, 14] which impacts performance. Furthermore, current differentiable rasterizertion methods fail to support differentiation with respect to many informative scene properties, such as textures and lighting, leading to low fidelity rendering, and less informative learning signals [14, 20, 21] .

---

In this paper, we present DIB-R, an approach to differentiable rendering, which, by viewing rasterization as a combination of local interpolation and global aggregation, allows for the gradients of this process to be computed analytically over the entire image. When performing rasterization of a foreground pixel, similar to [4], we define its value as a weighted interpolation of the relevant vertex attributes of the foreground face which encloses it. To better capture shape and occlusion information in learning settings we define the rasterization of background pixels through a distance-based aggregation of global face information. With this definition the gradients of produced images can be passed back through a variety of vertex shaders, and computed with respect to all influencing vertex attributes such as positions, colors, texture, light; as well as camera positions. Our differentiable rasterization's design further permits the inclusion of several well known lighting models.

We wrap our DIB-R around a simple neural network in which the properties of an initial polygon sphere are predicted with respect to some conditioning input. We showcase this framework in a number of challenging machine learning applications focusing on 3D shape and texture recovery, across which we achieve both numerical and visual state-of-the art results.

## 2  Related work

**Differentiable Rasterization:**   OpenDR [22], the first in the series of differentiable rasterization-based renderers, approximates gradients with respect to pixel positions using first-order Taylor approximation, and uses automatic differentiation to back-propagate through the user-specified forward rendering program. In this approach, gradients are non-zero only in a small band around the edges of the mesh faces, which is bound to affect performance. [14] hand-designs an approximate gradient definition for the movement of faces across image pixels. The use of approximated gradients, and lack of full color information results in noisy 3D predictions, without concave surface features. To analytically compute gradients, Paparazzi [18] and [19], propose to back-propagate the image gradients to the face normals, and then pass them to vertex positions via chain rule. However, their gradient computation is limited to a particular lighting model (Spherical Harmonics), and the use of face normals further prevents their approach to be applied to smooth shading. [25] designs a $C^\infty$ smooth differetiable renderer for estimating 3D geometry, while neglecting lighting and texture. [31] supports per-vertex color and approximates the gradient near boundary with blurring, which produces wired effects and can not cover the full image. [11] focus on rendering of point cloud and adopts a differentiable reprojection loss to constrain the distribution of predited point clouds, which loses point connectivity and cannot handle texture and lighting.

SoftRas-Mesh recently proposed in [20] introduces a probabilistic formulation of rasterization, where each pixel is softly assigned to *all* faces of the mesh. While inducing a higher computational cost, this clever trick allows gradients to be computed analytically. Parallel to our work, SoftRas-Color [21] extended this framework to incorporate vertex colors and support texture and lighting theoretically. However, in [21] each pixel would be influenced by all the faces and thus might result into more blurry predictions. The key difference between the parallel work of [21] and ours is that, similarly to [4], we specify each foreground pixel to the most front face and compute analytic gradients of foreground pixels by viewing rasterization as interpolation of *local* mesh properties. This allows our rendering effect the same as OpenGL pipeline and naturally supports optimization with respect to all vertex attributes, and additionally enables the extension of our pipeline to a variety of different lighting models. In contrast to [4], which also uses an interpolation-based approach, but applied to the entire image, our rasterization module allows for soft assignment of background pixels through an aggregation of global features.

**Adverserial 3D Object Generation:**   Generation of 3D shapes through deep learning has been approached using a Generative Adverserial Network (GAN) [5] in a plethora of work  [39, 1, 37, 30]. While these approaches require full 3D supervision, differentiable rendering frameworks allow learning 3D object distributions using only 2D supervision [10]. We showcase our model in the same application, where we are the first to learn a generator for both shape and texture.

## 3  DIB-R: Differentiable Interpolation-based Renderer

In this section, we introduce our DIB-R. Treating foreground rasterization as an interpolation of vertex attributes allows realistic images to be produced, whose gradients can be fully back-propagated through all predicted vertex attributes, while defining background rasterization as an aggregation of global information during learning allows for better understanding of shape and occlusion.

## 3.1 Rendering Pipeline

Many popular rendering APIs, such as OpenGL [36] and DirectX3D [23], decompose the process of rendering 3D scenes into a set of sequential user-defined programs, referred to as *shaders*. While there exist many different shader types, the vertex, rasterization, and fragment shaders the three most important steps for establishing a complete rendering pipeline. When rendering an image from a 3D polygon mesh, first, the vertex shader projects each 3D vertex in the scene onto the defined 2D image plane. Rasterization is then used to determine which pixels are covered and in what manner, by the primitives these vertices define. Finally, the fragment shader computes how each pixel is colored by the primitives which cover it.

The vertex and fragment shaders can easily be defined such that they are entirely differentiable. By projecting 3D points onto the 2D image plane by multiplying with the corresponding 3D model, view and projection matrices, the vertex shader operation is directly differentiable. In the fragment shader, pixel colors are decided by a combination of local properties including assigned vertex colors, textures, material properties, and lighting. While the processes through which this information are combined can vary with respect to the chosen rendering model, in most cases this can be accomplished through the application of fully differentiable arithmetic operations. All that remains for our rendering pipeline is the rasterization shader, which presents the main challenge, due to the inherently non-differentiable operations which it requires. In the following section we describe our method for rasterizing scenes such that the derivatives of this operation can be analytically determined.

## 3.2 Differentiable Rasterization

Consider first only the **foreground pixels** that are covered by one or more faces. Here, in contrast to standard rendering, where a pixel's value is assigned from the closest face that covers it, we treat foreground rasterization as an interpolation of vertex attributes[4]. For every foreground pixel we perform a z-buffering test [6], and assign it to the closest covering face. Each pixel is influenced exclusively by this face. Shown in Fig. 1, a pixel at position $\vec{p}_i$ is covered by face $f_j$ with three vertices $\vec{v}_0, \vec{v}_1, \vec{v}_2$, and each vertex has its own attributes: $u_0, u_1, u_2$, respectively. $\vec{p}_i$ and $\vec{v}_i$ are 2D coordinates on the image plane while $u_i$ are scalars. We compute the value of this pixel, $I_i$, using barycentric interpolation of the face's vertex attributes:

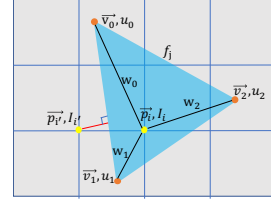

Figure 1: Illustration of our Differentiable Rasterization.

$$I_i = w_0 u_0 + w_1 u_1 + w_2 u_2, \tag{1}$$

where weights $w_0$, $w_1$ and $w_2$ are calculated over the vertex and pixel positions using a differentiable functions $\Omega_k$ (provided in Appendix):

$$w_k = \Omega_k(\vec{v}_0, \vec{v}_1, \vec{v}_2, \vec{p}_i), \quad k = 0, 1, 2. \tag{2}$$

While barycentric interpolation has been widely used in OpenGL pipeline. Here, we derive the differentiable reformulation. With this approach, it is easy to back-propagate gradients from a loss function $L$, defined on the output image, through pixel value $I_i$ to vertex attributes $u_k$ via chain rule:

$$\frac{\partial I_i}{\partial u_k} = w_k, \quad \frac{\partial I_i}{\partial \vec{v}_k} = \sum_{m=0}^{2} \frac{\partial I_i}{\partial w_m} \frac{\partial \Omega_m}{\partial \vec{v}_k}, \tag{3}$$

$$\frac{\partial L}{\partial u_k} = \sum_{i=1}^{N} \frac{\partial L}{\partial I_i} \frac{\partial I_i}{\partial u_k}, \quad \frac{\partial L}{\partial \vec{v}_k} = \sum_{i=1}^{N} \frac{\partial L}{\partial I_i} \frac{\partial I_i}{\partial \vec{v}_k}, \tag{4}$$

where $N$ is the number of pixels covered by the face. Now consider pixels which no faces cover, which we refer to as **background pixels**. Notice that in the formulation above, the gradients from background pixels cannot back-propagate to any mesh attributes. However, the background pixels provide a strong constraint on the 3D shape, and thus the gradient from them provide a useful signal when learning geometry. Take, for example, pixel $p_{i'}$ at position $\vec{p}_{i'}$ which lies outside of face $f_j$, in Fig 1. We want this pixel to still provide a useful learning signal. In addition, information from occluded faces an entirely ignored despite their potential future influence.

Inspired by the silhouette rasterizetion of [20], we define a distance-related probability $A_{i'}^j$, that softly assigns face $f_j$ to pixel $p_{i'}$ as:

$$A_{i'}^j = \exp(-\frac{d(p_{i'}, f_j)}{\delta}), \tag{5}$$

where $d(p_{i'}, f_j)$ is the distance function from pixel $p_{i'}$ to face $f_j$ in the projected 2D space, and $\delta$ is a hyper-parameter that controls the smoothness of the probability (details provided in Appendix). We then combine the probabilistic influence of all faces on a particular pixel in the following way:

$$A_{i'} = 1 - \prod_{j=1}^{n}(1 - A_{i'}^{j}). \tag{6}$$

where $n$ is the number of all the faces. The combination of all $A_{i'}$ into their respective pixel positions makes up our alpha channel prediction. With definition, any background pixel can pass its gradients back to positions of all the faces (including those ignored in the foreground pixels due to occlusion) with influence proportional to the distance between them in alpha channel. As all foreground pixels have a minimum distance to some face of $0$, they must receive an alpha value of $1$, and so gradients will only be passed back through the colour channels as defined above.

In summary, foreground pixels, which are covered by a specific face, back-propagate gradients though interpolation while background pixels, which are not covered by any face, softly back propagate gradients to all the faces based on distance. In this way, we can analytically determine the gradients for all aspects of our rasterization process and have achieved a fully differentiable rendering pipeline.

### 3.3 Rendering Models

In Equation 1, we define pixel values $I_i$ by the interpolation of abstract vertex attributes $u_0$, $u_1$ and $u_2$. As our renderer expects a mesh input, vertex position is naturally one such attribute, but we also simultaneously support a large array of other vertex attributes as shown in the Appendix. In the following section we outline the vertex attributes the rasterization can interpolate over and then back-propagate through, and the rendering models which the support of these attributes allows. A complete overview of this information is shown in Appendix.

#### 3.3.1 Basic Models

Our DIB-R supports basic rendering models where we draw the image directly with either vertex colors or texture. To define the basic colours of the mesh we support vertex attributes as either vertex colour or u,v coordinates over a learned or predefined texture map. Pixel values are determined through bi-linear interpolation of the vertex colours, or projected texture coordinates, respectively.

#### 3.3.2 Lighting Models

We also support 3 different local illumination models: Phong [26], Lambertian [16] and Spherical Harmonics [27], where the lighting effect is related to normal, light and eye directions.

To unify all the different lighting models, we decompose image color, $I$, into a combination of mesh color $I_c$ and lighting factors $I_l$ and $I_s$:

$$I = I_l I_c + I_s. \tag{7}$$

$I_c$ denotes the interpolated vertex colour or texture map values extracted directly from the vertex attributes without any lighting effect, $I_l$ and $I_s$ donate the lighting factors decided by specific lighting model chosen, where $I_l$ will be merged with mesh colour and $I_s$ is additional lighting effect that does not rely on $I_c$. We first interpolate light-related attributes such as normals, light or eye directions in rasterization, then apply different lighting models in our fragment shader.

**Phong and Lambertian Models:** In the Phong Model, image colour $I$ is decided by vertex normals, light directions, eye directions and material properties through the following equations:

$$I_l = k_d(\vec{L} \cdot \vec{N}), \qquad I_s = k_s(\vec{R} \cdot \vec{V})^{\alpha}, \tag{8}$$

where, $k_d$, $k_s$ and $\alpha$ are: diffuse reflection, specular reflection, and shininess constants. $\vec{L}$, $\vec{N}$, $\vec{V}$ and $\vec{R}$ are directions of light, normal, eye and reflectance, respectively, which are all interpolated vertex attributes. This results in the following definition for image colour under the Phong model:

$$I_{Phong} = I_c k_d(\vec{L} \cdot \vec{N}) + k_s(\vec{R} \cdot \vec{V})^{\alpha}. \tag{9}$$

As a slight simplification of full Phong shading we do not adopt ambient light and set light colour at a constant value of $1$. The Lambertian model can be viewed as a further simplification of the Phong Model, where we only consider diffuse reflection, and set $I_s$ as zero:

$$I_{Lambertian} = I_c k_d(\vec{L} \cdot \vec{N}). \tag{10}$$

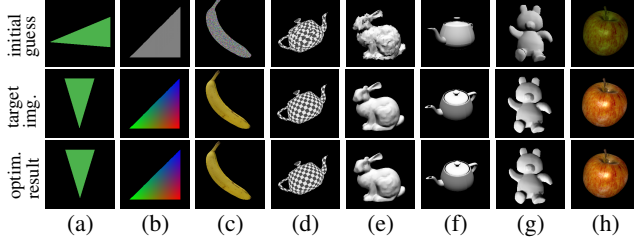

**Figure 2:** We perform a sanity check for our DIB-R by running optimization over several mesh attributes. We optimize w.r.t. different attributes in different rendering models: (a,b) vertex position and color in the vertex color rendering model, (c,d) texture and texture coordinates in the texture rendering model, (e,f) vertex and camera position in Lambertian model, (g) lighting in the Spherical Harmonic model, (h) material in the Phong model.

**Spherical Harmonic Model:** Here, $I_l$ is determined by normals while $I_s$ is set to 0:

$$I_{SphericalHarmonic} = I_c \sum_{l=0}^{n-1} \sum_{m=-l}^{l} w_l^m Y_l^m(\vec{N}),$$ (11)

where $Y_m^l$ is an orthonormal basis for spherical functions analogous to a Fourier series where $l$ is the frequency and $w_l^m$ is the corresponding coefficient for the specific basis. To be specific, we set $l$ as 3 and thus predict 9 coefficients in total. By adjusting different $w_l^m$, different lighting effects are simulated. For more details please refer to [27], Section 2.

**Optimization Results.** The design of our differentiable renderer allows for optimization over all defined vertex attributes and a variety of rendering models, which we perform a sanity check for in Fig. 2. Here, we optimize the L-1 loss between the target images (second row) and predicted rendered images. Note that [17] should be strictly better than us since it supports ray tracing. However, among rasterization-based renderers we are the first to support optimization of all vertex attributes.

## 4 Applications of DIB-R

We demonstrate the effectiveness of our framework through three challenging ML applications.

### 4.1 Predicting 3D Objects from Single Images

**Geometry and color:** We first apply our approach to the task of predicting a 3D mesh from a single image using only 2D supervision. Taking as input a single RGBA image, with RGB values $I$ and alpha values $S$, a Convolutional Neural Network $F$, parameterized by learnable weights $\theta$, predicts the position and color value for each vertex in a mesh with a specified topology (sphere in our case). We then use a renderer $R$ (specified by shader functions $\Theta$) to render the mesh predicted by $F(I, S; \theta)$ to a 2D silhouette $\tilde{S}$ and the colored image $\tilde{I}$. This prediction pipeline and the architecture details for our mesh prediction network $F$ are provided in Appendix.

When training this system we separate our losses with respect to the silhouette prediction, $\tilde{S}$, and the color prediction, $\tilde{I}$. We use an Intersection-Over-Union (IOU) loss for the silhouette prediction[2]:

$$L_{IOU}(\theta) = \mathbb{E}_{\mathbb{I}} \left[ 1 - \frac{||S \odot \tilde{S}||_1}{||S + \tilde{S} - S \odot \tilde{S}||_1} \right],$$ (12)

where $\odot$ denotes element-wise product. Note that $\{\tilde{I}, \tilde{S}\} = R(F(I, S; \theta))$ depend on the network's parameters $\theta$ via our DIB-R. We further use an L-1 loss for the colored image:

$$L_{col}(\theta) = \mathbb{E}_{\mathbb{I}} \left[ ||I - \tilde{I}||_1 \right].$$ (13)

When rendering our predicted mesh, we not only use the ground truth camera positions and compare against the original image, but also render from a random second view and compare against the ground truth renderings from this new view [14, 20]. This multi-view loss ensures that the network does not only concentrate on the mesh properties in the known perspective. We also regularize the mesh prediction with a smoothness loss [14, 20], $L_{sm}$, and a Laplacian loss [20, 29, 33], $L_{lap}$, which penalize the difference in normals for neighboring faces and the change in relative positions of neighboring vertices, respectively. The full explanation of these regularizers is provided in Appendix. The final loss function is then a weighted sum of these four losses:

$$L_1 = L_{IOU} + \lambda_{col} L_{col} + \lambda_{sm} L_{sm} + \lambda_{lap} L_{lap}.$$ (14)

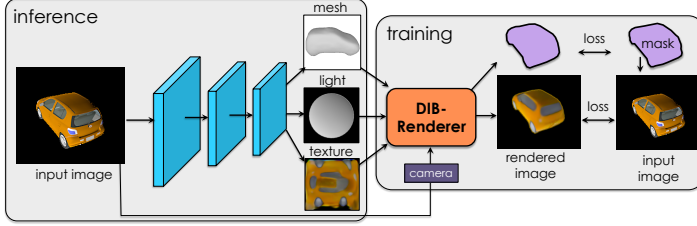

Figure 3: Full architecture of our approach. Given an input image, we predict geometry, texture and lighting. During training we render the prediction with a known camera. We use 2D image loss between input image and rendered prediction to to train our prediction networks. Note that the prediction can vary in different rendering models, e.g. texture can be vertex color or a texture map while the lighting can be Lambertian, Phong or Spherical Harmonics.

**Geometry, Texture, and Light.** We next apply our method to an extension of the previous task, where a texture map is predicted instead of vertex colors, and lighting parameters are regressed to produce higher quality mesh predictions. Our neural network $F$ is modified to predict vertex positions, a texture map, and various lighting information, depending on the lighting model used. Our full learning pipeline is shown in Fig. 3 and more details are included in the Appendix. We apply the same losses as in the previous section.

To increase the photo-realism of our predictions, we also leverage an adversarial framework [5, 24]. This is accomplished by training a discriminator network, $D(\phi)$, to differentiate between real images, $I$, and rendered mesh predictions, $\tilde{I}$, while our prediction network, $F$, simultaneously learns to make these predictions. We adopt the W-GAN [2] loss formulation with gradient penalty as in [7]:

$$L_{adv}(\theta, \phi) = \mathbb{E}_{\mathbb{I}}\left[ D(I; \phi) - D(\tilde{I}; \phi) \right], \quad L_{gp}(\phi) = \mathbb{E}_{\tilde{\mathbb{I}}}\left[ (||\nabla_{\tilde{I}} D(\tilde{I}; \phi)||_2 - 1)^2 \right]. \quad (15)$$

Similar to [12, 34, 40], we additionally use a perceptual loss and discriminator feature matching loss to make training more stable:

$$L_{percep}(\phi) = \mathbb{E}_{\mathbb{I}}\left[ \sum_{i=1}^{M_V} \frac{1}{N_i^V} ||V^i(I) - V^i(\tilde{I})||_1 + \sum_{i=1}^{M_D} \frac{1}{N_i^D} ||D^i(I; \phi) - D^i(\tilde{I}; \phi)||_1 \right], \quad (16)$$

where $V^i$ denotes the $i$-th layer of a pre-trained VGG network, $V$, with $N_i^V$ elements, $D^i$ denotes the $i$-th layer in the discriminator $D$ with $N_i^D$ elements, and the numbers of layers in network $V$ and $D$ are $M_V$ and $M_D$, respectively. Our full objective function for this task is then:

$$\theta^*, \phi^* = \arg\min_\theta \left( \arg\max_\phi (\lambda_{adv} L_{adv} - \lambda_{gp} L_{gp}) + \lambda_{per} L_{percep} + L_1 \right). \quad (17)$$

### 4.2 3D GAN of Textured Shapes via 2D supervision

In our second application, we further demonstrate the power of our method by training a Generative Adversarial Network (GAN) [5] to produce 3D textured shapes using only 2D supervision. We train a network $F_{GAN}$ to predict vertex positions and a texture map, and exploit a discriminator $D(\phi)_I$ to differentiate between real images, and rendered predictions. The network $F_{GAN}$ is modified so as to take normally distributed noise as input, in place of an image.

While empirically the above GAN is able to recover accurate shapes, it fails to produce meaningful textures. We suspect that disentangling both shape and texture by an image-based discriminator is a hard learning task. To facilitate texture modeling, we train a second discriminator $D(\sigma)_t$, which operates over texture map predictions. However, as our dataset does not contain true texture maps which can be mapped onto a deformed sphere, for ground truth textures we instead use the textures produced from our network trained to predict texture and lighting from images (Sec 4.1). To produce these texture maps, we pass every image in our training set through our texture and lighting prediction network, and extract the predicted texture. Our second discriminator then learns to differentiate between textures generated by $F_{GAN}$, and the extracted learned textures. We train $F_{GAN}$ via W-GAN with gradient penalty [7], and use both discriminators to provide a learning signal.

## 5 Experiments
**Dataset:** As in [14, 20, 33], our dataset comprises 13 object categories from the ShapeNet dataset [3]. We use the same split of objects into our training and test set as [33]. We render each model from 24 different views to create our dataset of RGB images used for 2D supervision. To demonstrate the multiple rendering models which DIB-R supports, we render each image with 4 different rendering models: 1) basic model without lighting effects, 2) with Lambertian reflectance, 3) with Phong shading, and 4) with Spherical Harmonics. Further details are provided in the Appendix.

Table 1: Results on single image 3D object prediction reported with 3D IOU (%) / F-score (%).

| Category | Airplane | Bench | Dresser | Car | Chair | Display | Lamp | Speaker | Rifle | Sofa | Table | Phone | Vessel | Mean |
|---|---|---|---|---|---|---|---|---|---|---|---|---|---|---|
| N3MR [14] | **58.5/80.6** | 45.7/55.3 | 74.1/46.3 | 71.3/53.3 | 41.4/39.1 | 55.5/43.8 | 36.7/**46.4** | 67.4/35.0 | 55.7/**83.6** | 60.2/39.2 | 39.1/46.9 | **76.2/74.2** | 59.4/**66.9** | 57.0/54.7 |
| SoftR. [20] | 58.4/71.9 | 44.9/49.9 | 73.6/41.5 | 77.1/51.1 | 49.7/40.8 | 54.7/41.7 | 39.1/39.1 | 68.4/29.8 | **62.0**/82.8 | 63.6/39.3 | 45.3/37.1 | 75.5/68.6 | 58.9/55.4 | 59.3/49.9 |
| Ours | 57.0/75.7 | **49.8/55.6** | **76.3/52.2** | **78.8/53.6** | **52.7/44.7** | **58.8/46.4** | **40.3**/45.9 | **72.6/38.8** | 56.1/82.0 | **67.7/43.1** | **50.8/51.5** | 74.3/73.3 | **60.9**/63.2 | **61.2/55.8** |

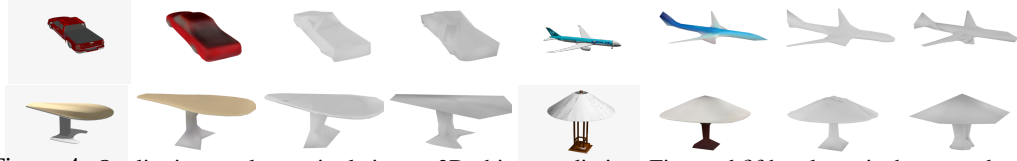

Figure 4: Qualitative results on single image 3D object prediction. First and fifth column is the ground-truth image, the second and sixth columns are the prediction from our model, the third and seventh column are results from SoftRas-Mesh [20], the rest two columns are results from N3MR [14].

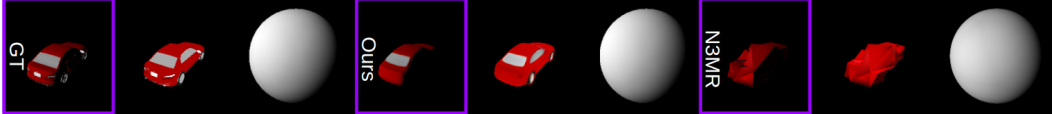

Figure 5: Qualitative examples for 3D shape, texture and light prediction. Col. 1-3: 1) GT rendered image with texture+light, 2) texture only rendered image, 3) light map. Col 4-6: our predictions. Col: 7-9: N3MR [14]

## 5.1 Predicting 3D Objects from Single Images: Geometry and Color

**Experimental settings:** In our experiments, we set $\lambda_{col} = 1, \lambda_{sm} = 0.001$, and $\lambda_{lap} = 0.01$. We train one network on all 13 categories. The network is optimized using the Adam optimizer [15], with $\alpha = 0.0001, \beta_1 = 0.9$, and $\beta_2 = 0.999$. The batch size is 64, and the dimension of input image is $64 \times 64$. We compare our method with the two most related differentiable renderers, N3MR [14] and SoftRas-Mesh [20], using the same network configurations, training data split and hyperparameters. For quantitative comparison, we first voxelize the predicted mesh into $32^3$ volume using a standard voxelization tool `binvox`[3] provided by ShapeNet [3] and then evaluate using 3D IOU, a standard metric in 3D reconstruction. We additionally measure the F-score following [32] between the predicted mesh and ground truth mesh. The tolerance for F-score is set to 0.02.

**Results:** Table 1 provides an evaluation. Our DIB-R significantly outperforms other methods, in almost all categories on both metrics. We surpass SoftRas-Mesh/N3MR with 1.92/4.23 points and 5.98/1.23 points in terms of 3D IOU and F-score, respectively. As the only difference in this experiment is the renderer, the quantitative results demonstrate the superior performance of our method. Qualitative examples are shown in Fig 4. Our DIB-R faithfully reconstructs both the fine-detailed color and the geometry of the 3D shape, compared to SoftRas-Mesh and N3MR.

## 5.2 Predicting 3D Objects from Single Images: Geometry, Texture and Light

**Experimental settings:** We adopt a UNet [28] architecture to predict texture maps. As we deform a mesh from a sphere template, similar to [13], we use 2D spherical coordinates as the UV coordinates. The dimension of the input image and predicted texture is $256 \times 256$. We use a 6-layer ResNet [9] architecture to regress the XYZ directions of light at each vertex from the features at the bottleneck layer of UNet network. We only use the Lambertian model and qualitatively compare with N3MR [14] by measuring the reconstruction error on rendered images under identical settings. Note that SoftRas-Mesh [20] does not support texture and lighting and so no comparison can be made. In the following sections, we only perform experiments on the car class, which has more diverse texture.

**Results:** We provide results in Table 2 and Fig 5. As ShapeNet does not provide ground-truth UV texture, we compute the L-1 difference on the rendered image using the predicted texture/texture+lighting and the GT image. Compared to N3MR [14], we achieve significantly better results both quantitatively and qualitatively. We obtain about 40% lower L-1 difference on texture and 60% smaller angle difference on lighting direction than N3MR. We also obtain significantly better visual results, in terms of the shape, texture and lighting, as shown in Fig 5 and the Appendix.

| Models | Texture | Lighting | Text. + Light |
|---|---|---|---|
| N3MR [14] | 0.03640 | 23.5585 | 0.02208 |
| Ours | **0.02179** | **9.7096** | **0.01362** |

Table 2: Results for texture and light prediction. Texture/Texture+Light shows L-1 loss on the rendered image for texture/texture+lighting. Lighting shows the angle between predicted lighting and GT lighting. Lower is better.

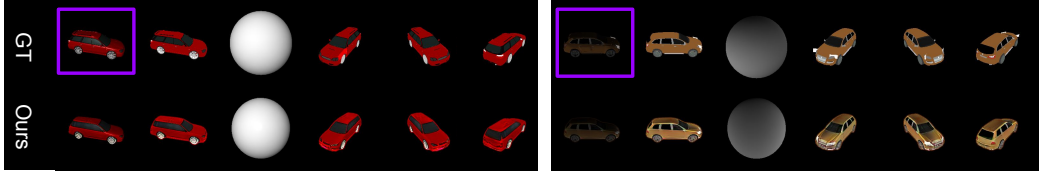

Figure 6: Qualitative examples for 3D shape, texture and light prediction, when exploiting **adverserial loss**. **Purple rectangle**: Input image. **Left Example**: Phong Lighting. **Right Example**: Spherical Harmonics. **First col**: Texture and Light. **Second col**: Texture. **Third col**: Light. **Forth to Sixth col**: Texture; different views.

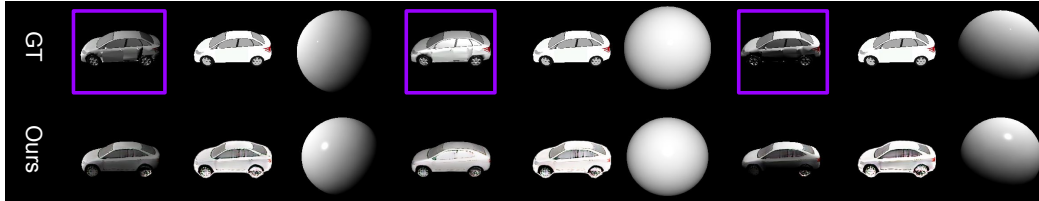

Figure 7: Light & Texture Separation Study. **Purple rectangle**: Input image, which are rendered with the same car model but different lighting directions. Each three columns visualize **Texture + Light, Texture, Light.**

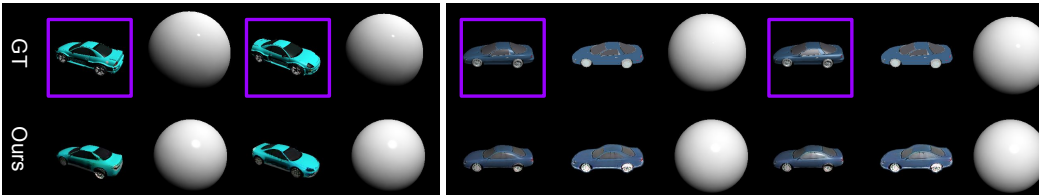

Figure 8: Light & Texture Separation Study. **Purple rect**: Input image. **Left:** Input images are with same light and texture but vary views. **Right:** Input images are with the same texture but with different shininess constants.

## 5.3 Texture and Lighting Results With Adversarial Loss

We now evaluate the effect of adding the adverserial loss to the previous experiment. We demonstrate our DIB-Render with Phong and Spherical Harmonic lighting models. For Phong model we keep diffuse and specular reflectance constant as 1 and 0.4 respectively and predict lighting direction together with shininess constant $\alpha$ while for Spherical Harmonic model we predict 9 coefficients.

**Experimental settings:** We first train the model without adversarial loss for 50000 iterations then fine-tune it with adversarial loss for extra 15000 steps. The detailed network architecture is provided in the Appendix. We set $\lambda_{adv} = 0.5, \lambda_{gp} = 0.5$, and $\lambda_{per} = 1$. We fix the learning rate for the discriminator to $1e^{-5}$ and optimize using Adam [15], with $\alpha = 0.0001, \beta_1 = 0.5$, and $\beta_2 = 0.999$.

**Qualitative Results and Separation Study:** We show qualitative results in Fig. 6. Notice that the network disentangles texture and light quite well, and produces accurate 3D shape. Furthermore, the adverserial loss helps in making the predicted texture look more crisp compared to Fig 5.

To further study texture and light disentanglement, we render test input images with the same car model but vary the lighting direction (Fig. 7). Our predictions recover the shape, texture map and lighting directions. Further examples are provided in Fig. 8. On the left, we fix the lighting and texture but render the car in different camera views, to illustrate consistency of prediction across viewpoints. On the right of the figure, we render images with different shininess constants, but fixed lighting direction, texture and camera view. Here, we find that our model is not able to accurately predict the shininess constant. In this case, the texture map erroneously compensates for the shininess effect. This might be because the shininess effect is not significant enough to be learned by a neural network though 2D supervision. We hope to resolve this issue in future work.

| Models | Texture | 2D IOU | Key Point |
|--------|---------|--------|-----------|
| CMR [13] | 0.041 | 0.262 | 0.930 |
| Ours | 0.041 | **0.243** | **0.971** |

Table 3: Results on CUB bird dataset [35]. Texture and 2D IOU show L-1 loss and 2D IOU loss between predictions and GT, lower is better. Key point evaluates percentage of predicted key points lying in the threshold of 0.1, higher is better.

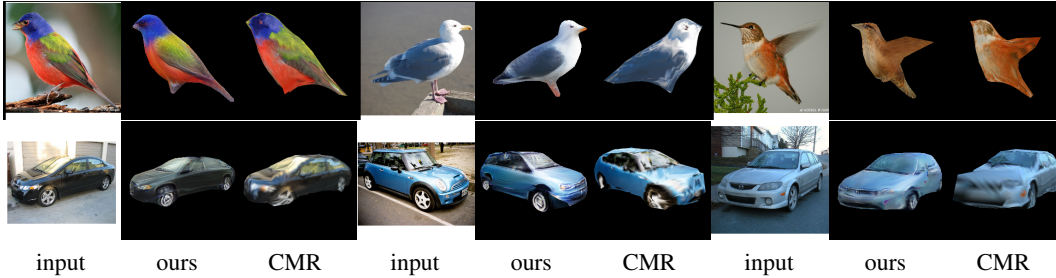

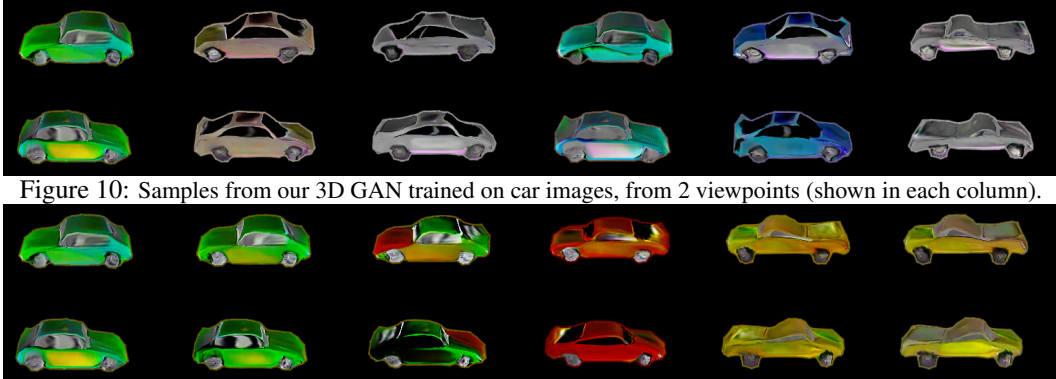

input      ours      CMR      input      ours      CMR      input      ours      CMR

Figure 9: Qualitative examples on CUBbird dataset [35] and PASCAL3D+ Car dataset [38]

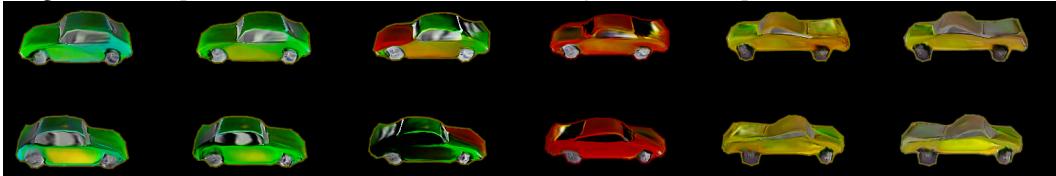

Figure 10: Samples from our 3D GAN trained on car images, from 2 viewpoints (shown in each column).

Figure 11: Renderings of objects produced by interpolating between latent codes of our 3D GAN, from 2 views

## 5.4 Real Images

**Experimental settings:** We next test our method on real images. Since real images generally do not have multiview captures for the same object, it would be very hard to infer precise light because light and texture would be entangled together. Following CMR [13], we adopt CUB bird dataset [35] and PASCAL3D+ car dataset [38], predicting shape and texture from a single view image.

**Results:** We compare our method with CMR [13]. Instead of predicting texture flow, we predict texture map directly. Both two methods use GT cameras estimated from structure from motion. Table 3 provides quantitative evaluation of predicted texture and shape on CUB bird dataset. Our show better shape and key points predictions than CMR. While the loss of texture predictions are the same, Fig. 9 shows qualitative improvements our method provides. Our textures are of higher fidelity and more realistic. This is because we predict a whole image as the texture map while CMR [13] adopts N3MR [14], which uses face color as the texture and so the restricted face size results in blurriness. For car prediction, both two methods clearly posses poor artifacts. This is because the segmentation in PASCAL3D+ car dataset is estimated from Mask R-CNN [8], which is not perfect. In addition, the car textures have more details than birds, which make it very hard to learn good shape and texture. Despite these facts, the visual quality our of predictions continues to display a marked improvement.

## 5.5 3D GAN of Textured Shapes via 2D Supervision

**Experimental settings:** We first train the networks to only predict shape, by only passing gradients back through the silhouette prediction. We then fix the shape prediction and only train to produce textures. We perform this experiment on the car class. Images from 4 primary views are rendered for each predicted mesh in each training iteration, and concatenated together when passed to rendered image discriminator to force generation of objects in a canonical pose.

**Results:** We show the results of randomly sampling from our learned distribution of car shapes and textures in Fig 10. This figure demonstrates the high quality of of shape and texture generations, in addition to their diversity. We also show the result of rendering meshes produced from interpolation between latent codes in Fig 11, to demonstrate the robust nature of our learned distribution.

## 6 Conclusion

In this paper, we proposed a complete rasterization-based differentiable renderer for which gradients can be computed analytically. Our framework, when wrapped around a neural network, learns to predict shape, texture, and light from single images. We further showcase our framework to learn a generator of 3D textured shapes.

**Acknowledgement** Wenzheng Chen wants to thank the support of DARPA under the REVEAL program and NSERC under the COHESA Strategic Network.

## Footnotes

[2]We denote $\mathbb{E}_{\mathbb{I}} \triangleq \mathbb{E}_{I \sim p_{data}(I)}$

[3] http://www.patrickmin.com/binvox/

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
