[Supplementary Material]

# Supplementary of Learning to Predict 3D Objects with an Interpolation-based Differentiable Renderer

**Wenzheng Chen**[1,2,3]   **Jun Gao**[1,2,3*]   **Huan Ling**[1,2,3*]   **Edward J. Smith**[1,4*]

**Jaakko Lehtinen**[1,5]   **Alec Jacobson**[2]   **Sanja Fidler**[1,2,3]

NVIDIA[1]   University of Toronto[2]   Vector Institute[3]   McGill University[4]   Aalto University[5]

{wenzchen, huling, jung, esmith, jlehtinen, sfidler}@nvidia.com, jacobson@cs.toronto.edu

## 1   Derivation of DIB-Render

In this section we show how to back propogate gradients from barycentric weights to the vertex positions via differentiable functions, $\Omega$. As shown in Fig. 1, the pixel at position $\vec{p}_i$ is covered by the face $f_j$ with three vertexes $\vec{v}_0$, $\vec{v}_1$, $\vec{v}_2$, $\vec{p}_i$ and $\vec{v}_i$ are 2D coordinates on the image plane and $w_i$ is corresponding barycentric weights. We derive $\Omega_0$, for wieght $w_0$ below, and weights $w_1$ and $w_2$ can be calculated similarly.

Figure 1: Illustration of our Differentiable Rasterization.

With the sum of barycentric weights being equal to 1, we have:

$$w_0 + w_1 + w_2 = 1. \tag{1}$$

We can rewrite it in matrix form as:

$$w_0 = 1 - \begin{bmatrix} 1 & 1 \end{bmatrix} \begin{bmatrix} w_1 \\ w_2 \end{bmatrix} \tag{2}$$

The Barycentric weights are calculated via pixel position $\vec{p}_i$ and face vertex positions $\vec{v}_0$, $\vec{v}_1$ and $\vec{v}_2$:

$$\vec{p}_i = w_0 \vec{v}_0 + w_1 \vec{v}_1 + w_2 \vec{v}_2, \tag{3}$$

where we can also rewrite it as:

$$\begin{bmatrix} \vec{v}_1 - \vec{v}_0 & \vec{v}_2 - \vec{v}_0 \end{bmatrix} \begin{bmatrix} w_1 \\ w_2 \end{bmatrix} = \begin{bmatrix} \vec{p}_i - \vec{v}_0 \end{bmatrix} \tag{4}$$

Thus we have:

$$\begin{bmatrix} w_1 \\ w_2 \end{bmatrix} = \begin{bmatrix} \vec{v}_1 - \vec{v}_0 & \vec{v}_2 - \vec{v}_0 \end{bmatrix}^{-1} \begin{bmatrix} \vec{p}_i - \vec{v}_0 \end{bmatrix} \tag{5}$$

If we merge Equation 2 and Equation 5, we can easily derive that:

$$w_0 = 1 - \begin{bmatrix} 1 & 1 \end{bmatrix} \begin{bmatrix} \vec{v}_1 - \vec{v}_0 & \vec{v}_2 - \vec{v}_0 \end{bmatrix}^{-1} \begin{bmatrix} \vec{p}_i - \vec{v}_0 \end{bmatrix} \tag{6}$$

In this way, $w_0$ can be treated as a output of a function while the input are pixel coordinate $\vec{p}_i$ and vertex positions $\vec{v}_0$, $\vec{v}_1$ and $\vec{v}_2$. We rewrite the weight $w_0$ as the $\Omega_0$ function and thus gradients can be back propagated from $w_0$ to vertex positions:

$$w_0 = \Omega_0(\vec{v}_0, \vec{v}_1, \vec{v}_2, \vec{p}_i) \tag{7}$$

---

Figure 2: Illustration of our rendering pipeline. (a) 3D mesh. (b) Rendered RGB image with white light in Lambertian model. (c) Rendered silhouette image with $\delta$ as 1.5e-4. (d) Rendered silhouette with $\delta$ as 1.5e-3. (e) Rendered silhouette with $\delta$ as 1.5e-5. (f) Rendered silhouette from SoftRas-Mesh[10]

| Vextex attribute | Vextex Shader | Rasterization | Fragment attribute | Fragment Shader |
|---|---|---|---|---|
| Vertices | | | | |
| Vertex Colors | Camera | | Pixel Colors | Color Model |
| Tex Coords | Model Matrix | Differentiable- | Pixel Tex Coords | Texture Model |
| Vertex Normals | View Matrix | Rasterizer | Pixel Normals | Lambertian Model |
| Light Directions | Projection Matrix | | Pixel Light Directions | Spherical harmonic Model |
| Eye Directions | | | Pixel Eye Directions | Phong Model |

Table 1: Differentiable vertex attributes and rendering models supported by our DIB-Render. With our method, we can differentiate most common attributes (Column 1 & 4, Vertex attributes and Fragment attributes) and apply it into multiple rendering models (Column 5, Rendering Models).

## 2   Rendering Pipeline

In this section we describe our rendering pipeline. Given a 3D mesh (Fig. 2, a), we render it into an RGBA image, where the RGB image (Fig 2, b, which is rendered with while light) is formed through our defined interpolation of texture or color information and the silhouette(alpha channel) (Fig. 2, c) is derived from our probabilistic distance function (See Eq. 5 & 6 in the paper).

$$d(p_{i'}, f_j) = \min_{p \in f_j} ||p_{i'} - p||_2^2 \qquad (8)$$

Inspired by SoftRas-Mesh [10], our distance function (Eq 8) also adopts square distance from the pixel $p_{i'}$ to the closest point in the face $f_j$. The image coordinate of $p_{i'}$ would be normalized into [-1, 1] to avoid bias from different image resolutions. There is a hyper-parameter, $\delta$ (See Eq.5 in the paper), to control the smoothness of the distance probability, where higher $\delta$ (Fig. 2, d) makes the silhouette blurry and lower $\delta$ (Fig. 2, e) will make the silhouette sharp. With this term we balance the information back propagated to the faces, and the sharpness of the predicted silhouette. In all of our experiments we set $\delta$ to 1.5e-4 (Fig. 2, c), as a middle ground between the two.

We also compare our defined silhouette definition with that of SoftRas-Mesh[10] (Fig. 2, f), where a sigmoid function is used to define their probabilistic distance function. However, the sigmoid function outputs 0.5 if a pixel lies on the edge of a face, leading many dim lines near face edges, and so provides a poorer learning signal when comparing to ground truth silhouettes.

## 3   Supported Mesh Attributes

In Table 1 we highlight the various rendering settings and scene attributes which our rendering pipeline supports. In Table 2, we compare supported properties in different methods.

| Model | Vertex Position | Vertex Color | Texture | Lighting | Analytic Grad. | Text. Coord. |
|---|---|---|---|---|---|---|
| OpenDR [12] | ✓ | ✓ | ✓ | ✓ | | |
| N3MR [6] | ✓ | * | ✓ | ✓ | | |
| Paparazze [8] | ✓ | * | * | * | ✓ | |
| SoftRas-Mesh [10] | ✓ | | | | ✓ | |
| SoftRas-Colour [11] | ✓ | ✓ | * | * | ✓ | |
| U3MM [2] | ✓ | ✓ | * | ✓ | ✓ | |
| Advgeo [9] | ✓ | ✓ | ✓ | ✓ | ✓ | |
| Ours | ✓ | ✓ | ✓ | ✓ | ✓ | ✓ |

Table 2: A comparison of different differentiable rasterization-based rendering models. ✓ means feature verified in the paper while * denotes feature supported in theory.

# 4 In-Depth Overview of Experiments

In this section, we provide the further details for each experiment in our paper.

## 4.1 Predicting 3D Objects from Single Images: Geometry and Color

**Dataset Details:** Following [6, 10, 18], we use 13 object categories from the ShapeNet dataset[2] [1], version 1. The training/testing split follows [18]. The training set contains 35007 objects, and test set contains 8752 different objects. For each object, we first normalize the object such that the center of the object is in the origin and all the vertexes lie in range $[-0.45, 0.45]$, we then render each object using Blender[3] with 24 different camera views. The camera views are equally distributed in a 360 degree ring around each object. The lighting direction in this dataset is set to uniform light. This results in 840k images for training and 210k images for testing.

**Network Structure:** We adopt the encoder-decoder framework as in [6, 10]. The encoder contains 3 convolutional layers and 3 linear layers, each convolutional layer has 64/128/256 channels with kernel size 5 and stride length of 2. The output feature map of the convolutional layers is flatten to $16384\text{-}d$ vector and fed to the first linear layer. Each linear layer has 1024 neurons. We add BatchNorm [4] to all the convolutional layers and the first and second linear layers. ReLu [7] activation function is used after each layer. We have two decoders, one for vertex position, the other for vertex color. Each decoder has three fully-connected layers, each layer has 1024/2048/1926 neurons, respectively. We directly predict position and color for each vertex. The template sphere has 632 vertexes and 1280 faces. We train every model until converge for around 2 days in V100 GPU.

**Loss Function:** The IOU loss and color loss has been described in the main paper, here we provide an explanation of the smoothness and Laplacian losses. We use the same smoothness loss design as in [6, 10]. Let $\mathbf{E}$ be the set of all edges, and $\theta_i$ be the angle between two neighboring faces, which share the edge $e_i$; the smoothness loss is then:

$$L_{sm} = \sum_{e_i \in \mathbf{E}} (\cos(\theta_i) + 1)^2, \tag{9}$$

which regularizes neighboring faces to have the similar normal directions, encouraging a smoother predicted mesh. Our Laplacian loss design follows [18]. For each vertex $v$, let $\mathcal{N}(v)$ be the neighboring vertices of $v$, then the Laplacian loss is defined as:

$$L_{lap} = (\delta_v - \frac{1}{||\mathcal{N}(v)||} \sum_{v' \in \mathcal{N}(v)} \delta_{v'})^2, \tag{10}$$

where $\delta_v$ is the predicted movement of vertex $v$. The Laplacian loss forces neighborhoods to move consistently [18].

**Metrics** We measure performance using 3D IOU loss and an F-score. The 3D IOU loss has been described in the main paper, and we explain the F-score here. We first uniformly sample 2500 points from the predicted mesh and ground truth mesh. For every predicted point, if the minimum distance from it to any ground truth point is less than a threshold, we treat it as a true positive, otherwise, it is false positive. This allows us to compute a precision score, $p$. For every ground truth point, if the minimum distance from it to every predicted point is less then a threshold, we treat this as a true positive, otherwise, it is a false negative, and so we can compute a recall score $r$. The F-score is then computed via: $2 * (p * r)/(p + r)$. We set the threshold to 0.02 in our experiments.

Figure 3: Qualitative comparison of texture and lighting with N3MR [6]. **Purple rectangle**: Input image. **First Row**: Ground Truth. **Second Row**: Prediction with DIB-Render. **Third Row**: Prediction of N3MR. **First column**: Texture and Light. **Second column**: Texture. **Third column**: Light

## 4.2 Predicting 3D Objects from Single Images: Geometry, Texture and Lighting

**Dataset Details** To learn diverse texture maps, we select to learn with the car category due to its large diversity of texture. In the experiments where we compare with N3MR, we choose Lambertian reflectance model as our rendering model since N3MR only supports such one lighting model. Instead of Blender, We choose to use OpenGL[4] to render data since we could easily turn on or turn off light effect in OpenGL shaders to separate the texture and light. We render car models into images both with and without light effect in Lambertian model and randomly sample the lighting direction over a quarter sphere where we constrain the light illuminates the upper part of the objects.

Similarly, for the experiment with adversarial Loss, since we adopting Phong and Spherical Harmonic lighting models in our learning framework, we use OpenGL render to render images with Phong and Spherical Harmonic lighting models respectively and train the neural network with the corresponding dataset.

**Network Structure** We adopt a UNet [16] model for texture prediction, the network architecture is similar to [5], except we add Batch Normalization to every convolutional layer and we use skip connections. We also replace all ReLu layers in UNet [16] to be Leaky Relu [20]. We exploit a shallow fully connected encoder for lighting parameters prediction. The encoder contains 1 linear input layer and 2 1D-residual linear layers [3] and 1 linear output layer. Output image feature for first three linear layers contains 1024/1024/1024 channels. For Phong model, the final linear layer outputs 4 lighting parameters where we try to learn the light direction together with the material shininess

Figure 4: Qualitative comparison of texture and lighting for model trained with and without adversarial loss. **Purple rectangle**: Input image. **First Row**: Ground Truth. **Second Row**: Prediction with adversarial loss. **Third Row**: Prediction without adversarial loss. **First column**: Texture and Light. **Second column**: Texture. **Third column**: Light. **Forth to Ninth column**: Texture from different views.

constant. For Spherical Harmonic model, the final linear layer outputs 9 coefficients of Spherical Harmonic basis.

For experiments with adversarial loss, we inherit basic discriminator architecture from DC-GAN [15]. We use Instance normalization [17], and Leaky-ReLU [13].

### 4.3 Real Images

**Dataset Details & Network Structure**   We adopt the same dataset and network structure with CMR [5] but change the differentiable renderer from N3MR [6] to our method. For more details please refer to CMR [5].

### 4.4 3D GAN

**Dataset Details**   We use the car class from the ShapeNet dataset for this task[1]. We render each car from 4 primary viewpoints: front, back, left and right. In addition to these images we learn a dataset of car textures using our texture and lighting prediction model. For each car object we pass one rendered view through our learned texture predictor and save the result as an example 'ground truth' texture.

**Network Structure**   Four networks are used in this application: a shape generator $G_1$, a texture generator, $G_2$, a rendered image discriminator, $D_1$, and a texture discriminator, $D_2$. $G_1$ expects a vector of random noise of length 128, and passes it through a series of 8 fully connected layers with ReLU activations [14], and batch normalization [4] to steadily increase its dimension to 1926 ($3 \times$ time number of vertices). It is then reshaped for output as the predicted positions for each vertex. In addition, activations from its fifth layer are outputted as conditioning information for the texture prediction. This information is concatenated with a second vector of random noise, also of size 128, and passed to the Texture Generator $G_2$. The texture predictor is comprised of 7 convolutional layers, again with ReLU activations and batch normalization, to output a texture map with 256 by 256 resolution. The image discriminator, $D_1$, takes as input a 64 by 64 resolution image, rendered from our generated mesh. Its architecture is composed of 4 convolutional layers with leaky-ReLU

activations [13], and instance normalization [17], followed by a single fully connected layer to output a vector of length 32. An identical architecture is leveraged for $D_2$, except the first layer is modified to accept the larger texture resolution of 256 by 256.

## 5 More Results

### 5.1 Single Image 3D Reconstruction

We provide more results on more different categories and compare with Softras-Mesh [10] and N3MR [6] in Figure 6. We further provide more results from different view angles on Figure 7, demonstrating the high fidelity mesh our model could predict.

### 5.2 Predicting 3D Objects from Single Images: Geometry, Texture and Light

We first show qualitative comparison with N3MR[6] in Fig. 3, then we show qualitative comparison for model trained with and without adversarial loss in Fig.4. We show more qualitative results with adversarial loss in Fig. 5. We further show more separation study cases in Fig. 8 and Fig. 9.

### 5.3 Real Images

In Figure 10 we show more examples of the learned shape and texture from CUB bird dataset [19]. Our method successfully learns the hard shapes such as beaks, legs and wings together with realistic textures.

### 5.4 3D GAN

In Figure 11 we show examples of the learned texture we train our texture discriminator. Figure 12 and Figure 14 show a random sample of textures and meshes produced from our mesh generators. Here, we demonstrate that our method is able to properly learn the target distribution of obejcts, both in terms of quality and diversity. In Figure 13 and Figure 15 we show textures and meshes produced when interpolating between random latent vectors passed to our mesh generators. This interpolation highlights that our learned distribution is robust, such that sampling along a line in latent space produces high fidelity textures at meshes every step.

## Footnotes

[2]www.shapenet.org

[3]https://www.blender.org/

[4]https://www.opengl.org/

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

Figure 5: Qualitative examples for 3D shape, texture and light prediction. **Purple rectangle**: Input image. **First to fourth rows**: Phong Model. **Fifth to seventh rows**: Spherical Harmonic Model. **First column**: Texture and Light. **Second column**: Texture. **Third column**: Light. **Forth to Ninth column**: Texture from different views.

Figure 6: Qualitative comparisons on single image 3D object prediction. First column is the ground-truth image, the second and third columns are the prediction from our model, the forth and fifth column are results from SoftRas-Mesh [10], the last two columns are results from N3MR [6].

Figure 7: Qualitative results on single image 3D object prediction from different views. The first column is the ground truth, the rest columns are images where we rotate predictions with different angles.

Figure 8: Light & Texture Separation Study. **Purple rectangle**: Input image, which are rendered with the same car model but different lighting directions. Each three columns visualize **Texture + Light, Light, Texture.**

Figure 9: Light & Texture Separation Study. **Purple rect**: Input image. Input images are with same light and texture but vary views. Each three columns visualize **Texture + Light, Light.**

Figure 10: Qualitative results on CUB bird dataset [19]. The first row shows GT image and mask, Our predictions and CMR [5] predictions, respectively. The second row shows the learned shape and texture rendered in multiple views.

Figure 11: Ground Truth Textures. Samples from the distribution of learned textures through which we train our texture discriminator.

Figure 12: GAN textures. Random sample of textures produced from our texture generator.

Figure 13: Interpolation of GAN Textures.Textures produced when interpolating between random latent vectors passed to our texture generator.

Figure 14: Rendered GAN samples. Renderings of objects produced by sampling from our mesh generators.

Figure 15: Interpolated GAN samples. Renderings of objects produced by interpolating between random latent vectors passed to our mesh generators.