[Reviews · NeurIPS 2019]

Reviewer 1



The paper is clearly and well written and together with the supplement it should be possible to reimplement the renderer and the experiments. The qualitative experiments on synthetic data are impressive and the quantitative evaluation shows the performance of the proposed system relative to others. Developing differentiable rewnderers is important to enable the inference of 3D quantities such as geometry, and lighting effects based on 2D image observations. Since rasterization is still one of the most used rendering techniques for various areas of research (excepting graphics) a differentiable model for this renderer is important to advance the state of the art in the community. The authors show that the proposed renderer can be used to take analytic gradients with respect to all commonly used image formation parameters. This is an important contribution. Some questions: 1) the renderer is called DIB but it seems the propper acronym would be IBD (since it seems to stand for Interpolation-based Differentiable). I would make that consistent. 2) Regarding z-buffering: It seems that in gradient updates to the vertex locations the z-buffering might change (i.e. which primitive is the closest one). Do you recompute the z-biuffering at each gradient step or is it fixed based on the initial view when optimizing vertex locations? 3) In Fig. 2 f) you seem to be optimizing over the pose of the tea pot. Is it simply a matter of taking gradients all the way down to the pose of the tea pot and updating that in a gradient descent fashion? 4) It would be useful to get an intiuition for relative timing between the proposed renderer and others to see that tradeoff and some more reasoning as to why one should us the rasterization based renderer vs the ray tracing one form [15]. Is it simplicity? Computational efficiency? Speed? 5) l 126: I am not sure what you mean b y the alpha channel prediction? Do you simply mean that you store the value of A_i' in the alpha channel a t pixel p_i' ?

Reviewer 2



This work introduce a differentiable renderer based on rasterization. Different from previous differentiable renderers, DIB-Renderer supports texture maps as well, which enables the existing 3D object reconstruction methods to predict the texture of the object. DIB-Render is based on rasterization. It interpolates the vertex's attributes for each foreground pixel. It also softly assign vertex to background pixel so that it can back-propagate mesh attributes. The renderer also supports three different lighting models: Phong, Lambertian and Spherical Harmonics. The author also shows different applications for the renderer such as single image 3D reconstruction and estimating geometry, texture and lighting condition. Extensive experiments prove the purposed renderer is superior to the existing differentiable renderers and achieves plausible results on the applications they purposed. The paper is well-written and structured clearly. The purposed renderer can be put into use for future 3D reconstruction tasks. The current 3D reconstruction put more emphasis on the geometry than textures partially due to the lack of a differentiable renderer that supports texture maps. This work could result in more realistic 3D reconstruction.

Reviewer 3



The paper describes a differentiable renderer for triangle meshes that provides gradients with respect to geometry, texture, and illumination. As a key innovation, the paper claims the formulation of rasterization as interpolation, saying that "In contrast to standard rendering, where a pixel’s value is assigned from the closest face that covers the pixel, we treat rasterization as an interpolation of vertex attributes". However, the formulation in Equation 1 using barycentric interpolation is textbook material in computer graphics. I do not think using this standard approach in a differentiable renderer can be claimed as a significant novel contribution. The experimental results on single image 3D reconstruction and textured shape generation using only 2D supervision lead to good quality, but they do not provide a significant improvement over previous work on these problems in my opinion. I consider the fact that this renderer also supports gradients due to illumination and texture, which some of the other public implementations don't (or they focus on specific cases, like spherical harmonics illumination), more as an engineering detail. The state of the art here is the work by Li et al. on differentiable ray tracing, which even supports gradients due to indirect illumination. The paper should also discuss "Pix2Vex: Image-to-Geometry Reconstruction using a Smooth Differentiable Renderer" by Petersen et al., and "Unsupervised 3D Shape Learning from Image Collections in the Wild" by Szabo and Favaro. Both use differentiable rendering, for single view shape reconstruction or shape generation using 2D supervision only similar as in this submission. In summary, the technical contribution in the differentiable renderer does not seem significant enough to me for a NeurIPS paper. Experiments are performed on standard problem statements (single image reconstruction, shape generation from 2D supervision) with good results, but previous work achieves quite similar quality.

Reviewer 4



-- The model -- The paper presents a differentiable renderer (DIB-Render) that can render a coloured 3D mesh onto a 2D image. Having such renderer allows, for example, to train a neural network that can reconstruct a 3D shape of an object from a single image and render the shape onto a number of 2D views using different camera configurations. The learning can then be supervised by computing a reconstruction error between the computed rendering of a 3D shape and an actual image (using an L1 loss for the coloured image or Intersection over Union (IoU) for the binary silhouettes). The renderer is largely based on the soft rasterizer (Soft-Ras) proposed in [18, 19]. Unlike traditional non-differentiable rasterizers, which assign a binary score of whether a pixel in the image plane is covered by a triangle or not, Soft-Ras computes a soft score based on a distance of a pixel to the triangle (with an exponential or a sigmoid function of distance). This allows to compute gradients of image pixels with respect to the vertex attributes such as coordinates, colours and so on. One noteworthy difference between the proposed renderer and Soft-Ras is that the former uses only the closest triangle ("For every such pixel we perform a z-buffering test [6], and assign it to the closest covering face"), whereas the latter uses a soft z-buffering formulation where the triangles behind the closest one can still receive gradients using SoftMin function. Since in [19] it was argued as an advantage, I would like to ask the authors what was the reason behind not using the soft z-buffering? Another minor difference is that the DIB-Render uses exponential function (Equation (5)), whereas Soft-Ras uses Sigmoid (Equation (1) in [19]). Now on what separates the proposed renderer from Soft-Ras. First, DIB-Render can sample pixel colours not only from the vertices but also from the textures in the UV coordinate space, and differentiate with respect to both texture coordinates and textures themselves. This allows to train a texture reconstruction network (a UNet type architecture) alongside with the mesh-reconstruction network. Secondly, DIB-Render supports different lighting models including Phong, Lambertian and Spherical Harmonics. Importantly, it allows to compute derivatives with respect to the lighting direction and train a network that predicts lighting. Thirdly, an adversarial loss applied to the generated 2D renderings in order to distinguish them with the actual images, as well as applied directly in the space of UV texture maps. It helps to improve the crispness of reconstructed textures. The fourth contribution is a 3D GAN, that at training time reconstructs a 3D mesh and a UV texture from a noise vector and, since no paired projections are available, is trained only with an adversarial loss. This, in theory, allows to lift the requirement of having multi-view images for training such models. -- Experiments -- - In the first experiment DIB-Render is compared with the Soft-Ras [18] and Neural Mesh 3D Renderer (N3MR) [12] on the task of Single-Image 3D shape reconstruction (Tab 2). DIB-Render outperforms the baselines on both IoU metric as well as on F-score between the predicted mesh and ground truth meshes. However, I have a concern about this experiment. As the lighting is not taken into account as in this experiment, the DIB-Render has very little differences to the SoftRas, and thus I expect them to perform about the same. It is important to understand, what makes DIB-Render better. Is it the differences in z-buffering or something else? - In the second experiment, DIB-Render is evaluated on the task of texture reconstruction and light direction prediction. DIB-Render outperforms N3MR both on the texture reconstruction accuracy and on the angular difference between the predicted and the actual lighting direction. - In the third experiment, using adversarial loss is evaluated for reconstruction of shape under Phong lighting model and Spherical Harmonics. Using adversarial loss enables to generate crisper textures. - In the fourth experiment the 3D GAN is evaluated. I was confused, because in the model section (Sec. 4.2) it is stated, that the "GAN is able to recover accurate shapes, but it fails to produce meaningful textures." (l.223). However in the experiments (Sec. 5.4) I see the opposite: "this figure demonstrates the high quality of of shape and texture generations" (l.299). At least, as far as I can see, generated textures are not very realistic, which could be explained by the lack of reconstruction loss (L1) with only adversarial loss being used. This conforms with the behavior of Pix2Pix architecture, which requires L1 loss to stabilise training and doesn't work very well without it. -- Some comments and criticism -- - It is worth to cite and discuss the differences to [a] that also uses a smooth differentiable mesh renderer and applies a variety of adversarial losses on the rendered shapes. Another paper [b] that uses a smooth differentiable renderer of point clouds is also worth mentioning in the related work. - Figures with qualitative results are very difficult to read and make sense of, because of the lacking row and column titles directly in the figures (Fig. 3, 5-9, Fig. 5 of the Appendix and so on). One has to repeatedly jump between reading the figure caption and the figure itself to understand what each column corresponds to. The figures need to be improved. - l.166 - I believe should be "light, normal, reflectance and eye, respectively", i.e. "R" is is reflectance direction and "V" is the viewer. [a] Pix2Vex: Image-to-Geometry Reconstruction using a Smooth Differentiable Renderer. Petersen et al. ArXiv:1903.11149, 2019. [b] Unsupervised Learning of Shape and Pose with Differentiable Point Clouds. Insafutdinov and Dosovitskiy, NeurIPS, 2018.

[Author Response · NeurIPS 2019]



Figure 1: Qualitative comparisons for prediction from real images. Top is CUB-bird dataset, bottom is PASCAL3D+ car dataset

Figure 2: Qualitative results on Airplane.

We would like to thank reviewers for their detailed comments. We have attempted to address all concerns below.

**Contribution [R4]** We wish to emphasize that although barycentric interpolation has been used before in a forward
rendering pipeline, it is non-differentiable due to rasterization, our contribution is its **differentiable reformulation**.
More importantly, while barycentric interpolation only affects foreground pixels, we further employ a global aggregation
method to obtain a probabilistic silhouette which handles background pixels, making DIB-Renderer a solution for all
the pixels in the image. To the best of our knowledge, DIB-Renderer is the first analytical differentiable renderer which
supports all common elements in a rendering pipeline, while also supporting optimization over large shape deformation.
We also want to clarify that supporting texture mapping and illumination is **not a trivial engineering problem**. SoftRas-
Mesh [18] and SoftRas-Color [19] do not support texture and lighting of this form. N3MR [12] only supports face
sampling based texture mapping and Lambertian illumination. However, this texture mapping samples the same number
of pixels per face, resulting in inaccuracy for large faces. Both models cannot be easily extended to support advanced
illumination and texture models due to specific design choices in their differentiable rendering process which would
require a major algorithm reformulation. In contrast, our formulation which, due to intentional similarities and parallels
to standard OpenGL texture mapping, naturally supports texture and lighting models.

**Additional Experiments [R2,R3,R4]** To further demonstrate the effectiveness of our DIB-Renderer, we show ad-
ditional 3D reconstruction results for real images and a new ShapeNet class. For **real images**, we follow Learning
Category-Specific Mesh Reconstruction from Image Collections (CMR [11]), and use two datasets: the CUB bird
dataset and the PASCAL3D+ car dataset. We predict geometry and texture from a single-view image. Results are shown
in Fig. 1 and Tab. 1, where we demonstrate more faithful geometry and more realistic textures. For **new ShapeNet
class**, examples on the Airplane class are shown in Fig. 2 and we will put more results in an updated supplementary.

| Models | Texture | 2D IOU | Key Point |
|--------|---------|--------|-----------|
| CMR [11] | 0.043 | 0.262 | 0.930 |
| Ours | 0.043 | **0.243** | **0.972** |

Table 1: Results on CUB bird dataset. Texture and 2D IOU show L-1 loss and 2D IOU loss between predictions and GT. (Lower is better) Key point evaluates percentage of predicted key points lying in the threshold of 0.1. (Higher is better)

**Related Work MCRT [R2,R4]** We agree that the Monte-Carlo Ray Tracing based method [15] supports more advanced
features. However, there are 2 main benefits to our method. Firstly, DIB-Renderer is faster in terms of running time
since we do not need to sample millions of rays. To demonstrate this, we render a 3D human model with 6890 vertices
and 13776 faces into a 256x256 image with a Titan-V GPU with the same camera settings and calculate the gradients
for all operations. We show the results of this comparison in Tab. 2, which shows a roughly 7.7x speed increase, over
MCRTs fastest optimization setting. Secondly, MCRT has not yet been demonstrated to work for machine learning
applications where the initialization does not already lie very close to the target.

| Ours | MCRT-4 | MCRT-16 | MCRT-64 | MCRT -256 |
|------|--------|---------|---------|-----------|
| **0.0234** | 0.1819 | 0.4485 | 1.6408 | 6.1288 |

Table 2: Running time (seconds) for one iteration (forward + backward). For MCRT, the more rays you sample, the higher running time it is. We are 7.7x faster than MCRT even when only 4 rays are sampled.

**Softras [R5]** We wish to emphasize that we treat Softras-Mesh [18] and Softras-Color [19] differently since they are
released in Jan, 2019 and April, 2019, respectivley. Thus, we view Softras-Color as concurrent work. Our treatment
of background pixels is inspired by Softras-Mesh, as stated in our paper. We agree that soft-z-buffer is a promising
direction for occlusion, however our choice of complete face aggregation for background pixels also implicitly handles
occlusion. In the first experiment (Sec. 5.1), we compare our method with Sofrtas-Mesh, which does not have a z-buffer.
For the difference between use of the exponential and sigmoid function, we show the comparisons in the supplement.
The exponential function is a better approximation of silhouettes, removing dim lines near face edges (Sec. 2 and Fig.
2), allowing our method to outperform Softras-Mesh on geometry prediction.

**Other mentioned papers [R3,R4,R5]** Petersen et al. estimates 3D geometry only and neglects lighting and texture.
Szabo et al. only supports per-vertex color and approximates the gradient near boundary with blurring, which produces
wired effects and can not cover the full image. Both produce objectively worse results than ours (Fig. 10 in Petersen
et al. and Fig. 8 in Szabo et al. v.s. Fig. 3,4,5 in our paper). Insafutdinov et al. focus on rendering of point cloud
and adopts a differentiable reprojection loss to constrain the distribution of predited point clouds, which loses point
connectivity and cannot handle texture and lighting. We will included the suggested citations in a revision.

**Explanation [R2,R3,R5]: R2. Z-buffer** We recompute the z-buffer for each iteration. **R2. Camera optimization in
Fig.2 (f)** We optimize camera parameters (rotation and translation matrices) via gradient descent, while fixing other
parameters (vertex, light). **R2. Alpha channel** Besides RGB channel, we create an alpha channel that stores $A_i$, which
represents the probability that pixel i is covered by the mesh. **R3. Quantitative results for lighting & texture** We
compare lighting and texture in Tab. 3 of our paper. **R5. 3D GAN** Line 223 is a straight-forward GANs formulation,
we have a second discriminator operating on texture map directly and this produces better results in line 299. **R5. Line
166 and figures** It should be "light, normal, reflectance and eye". We will improve the figures in a refined version.

[Meta-Review · NeurIPS 2019]

This paper received high-variance reviews, with some in favor, some borderline, and one against. Ultimately, the decision was made to accept this paper: a differentiable rasterizer that supports the full set of standard rendering features (lighting, texture, etc) in one package is a valuable contribution. The committee does have one reservation. The main contribution of the paper is a differentiable form of rendering based on Barycentric interpolation within triangles. The authors claim this as a novel algorithm, but this is actually a standard procedure used by virtually all rasterization-based renderers. The authors should tone down their claims of algorithmic novelty and explicitly acknowledge the strong connection to classic rasterization.